# Gene Enrichment Analysis of Astrocyte Subtypes in Psychiatric Disorders and Psychotropic Medication Datasets

**DOI:** 10.3390/cells11203315

**Published:** 2022-10-21

**Authors:** Xiaolu Zhang, Alyssa Wolfinger, Xiaojun Wu, Rawan Alnafisah, Ali Imami, Abdul-rizaq Hamoud, Anna Lundh, Vladimir Parpura, Robert E. McCullumsmith, Rammohan Shukla, Sinead M. O’Donovan

**Affiliations:** 1Department of Neurosciences, University of Toledo, Toledo, OH 43614, USA; 2Department of Neurobiology, The University of Alabama at Birmingham, Birmingham, AL 35294, USA; 3Promedica Neurosciences Institute, Toledo, OH 43606, USA

**Keywords:** astrocyte subtype, psychotropic medication, schizophrenia, major depression, kinase

## Abstract

Astrocytes have many important functions in the brain, but their roles in psychiatric disorders and their responses to psychotropic medications are still being elucidated. Here, we used gene enrichment analysis to assess the relationships between different astrocyte subtypes, psychiatric diseases, and psychotropic medications (antipsychotics, antidepressants and mood stabilizers). We also carried out qPCR analyses and “look-up” studies to assess the chronic effects of these drugs on astrocyte marker gene expression. Our bioinformatic analysis identified gene enrichment of different astrocyte subtypes in psychiatric disorders. The highest level of enrichment was found in schizophrenia, supporting a role for astrocytes in this disorder. We also found differential enrichment of astrocyte subtypes associated with specific biological processes, highlighting the complex responses of astrocytes under pathological conditions. Enrichment of protein phosphorylation in astrocytes and disease was confirmed by biochemical analysis. Analysis of LINCS chemical perturbagen gene signatures also found that kinase inhibitors were highly discordant with astrocyte-SCZ associated gene signatures. However, we found that common gene enrichment of different psychotropic medications and astrocyte subtypes was limited. These results were confirmed by “look-up” studies and qPCR analysis, which also reported little effect of psychotropic medications on common astrocyte marker gene expression, suggesting that astrocytes are not a primary target of these medications. Conversely, antipsychotic medication does affect astrocyte gene marker expression in postmortem schizophrenia brain tissue, supporting specific astrocyte responses in different pathological conditions. Overall, this study provides a unique view of astrocyte subtypes and the effect of medications on astrocytes in disease, which will contribute to our understanding of their role in psychiatric disorders and offers insights into targeting astrocytes therapeutically.

## 1. Introduction

Astrocytes play many important roles in the brain, from facilitating metabolism and cell signaling [1,2], to responding to injury through regulating oxidative stress [3]. However, the role of astrocytes in central nervous system (CNS) disorders is still poorly understood. While studies of postmortem brain broadly support perturbation of astrocytes in neurological and psychiatric disorders, the findings can prove challenging to interpret. For example, there is evidence of both increased [4,5,6,7,8] and decreased [7,9,10,11,12,13,14,15,16,17,18,19] astrocyte marker expression, increased [6,10] and decreased [20,21,22,23,24,25,26] astrocyte cell counts, changes in astrocyte morphology [7,27,28,29], as well as no significant changes in astrocyte expression [27,30,31,32,33,34,35] in disorders like schizophrenia and major depressive disorder (MDD). These findings also vary depending on the brain region and methodological approach studied [6,7,9,11,21,28,36,37,38,39]. A potential cause for these conflicting findings may be due to the complex responses of astrocytes to injury; astrocytes can undergo remodeling that leads to a range of responses [40] from complete atrophy [41,42] to a dynamic state termed “reactive astrogliosis” [43,44,45].

Reactive astrogliosis describes a spectrum of heterogeneous changes in gene expression [46,47,48], cell morphology [47,48,49,50,51,52], and overall function including fluid and ion homeostasis [53,54], oxidative stress response [55,56,57] and synapse formation [58,59]. These changes can also range from reversible to permanent [60,61]. A diverse set of mechanisms play a role in inducing astrogliosis [62], which can have a protective or pathological effect [63,64,65], depending on the subtype of astrocyte involved [43,46], and the type of injury sustained [43,49,62]. In mouse models of ischemic injury or lipopolysaccharide (LPS)-induced infection, changes in astrocyte gene expression were identified in a common set of genes and genes that were uniquely regulated by each type of injury [43]. There are also different “types” of astrocytes that can have potentially different responses to injury. These subtypes are classified based on cellular morphology (e.g., protoplasmic, fibrous, bushy), location (e.g., frontal cortex, hippocampus, cerebellum) and primary functions (e.g., metabolic, structural, signaling) [66]. However, our understanding of the different types of astrocytes in humans and their roles in CNS disorders is still limited [66,67]. Although inclusive of this wide spectrum of phenotypic changes, the term “reactive astrogliosis” will be used here as an umbrella term to encompass the range of possible changes that occur in astrocytes as a result of neurological insult.

In addition to disease heterogeneity and methodological differences, psychotropic medications are hypothesized to normalize disease-associated changes in astrocyte expression [27,66,68], and may contribute to the conflicting reports of altered astrocyte expression reported in psychiatric disorders. Antipsychotic treatment is associated with changes in brain volume in schizophrenia patients [69,70] and in nonhuman primates administered antipsychotic drugs [71,72]. Antidepressant use is associated with reduced astrocyte cell counts in rodent brain [73] and reduced astrocyte marker gene expression in postmortem MDD brain tissue studies [15], although other studies report no changes in these variables in postmortem studies [12,23,24,26,74]. Mood stabilizers are also associated with changes in astrocyte morphology and density in rodent brain [75]. Despite the extensive number of studies examining astrogliosis in postmortem and animal studies of neurological and psychiatric disorders, it is still unclear what effect these medications have on astrocyte expression and function in these disorders.

In this study we apply bioinformatic analysis of transcriptomic datasets to assess the relationship between different astrocyte subtypes (mouse astrocyte subtypes, mouse disease model astrocytes, human astrocytes, human SCZ hiPSC-derived astrocytes) and psychiatric disorders, and between astrocytes and psychotropic medications. Our goal is first to explore how different subtypes of astrocytes are implicated in psychiatric disorders and second, to determine whether psychotropic medications are associated with molecular changes in astrocytes in disease.

## 2. Materials and Methods

### 2.1. Mouse Astrocyte Subtype Gene-Sets

The gene-sets for each transcriptomically distinct subtype of mouse astrocyte were downloaded from the supplementary data of Batiuk et al. single-cell RNA sequencing astrocyte study [76]. Full gene-sets are listed in Appendix A. These gene-sets include genes enriched in each of five subtypes of astrocytes (AST1-5) and genes expressed in at least 60% of astrocytes (common) identified in the study. As described in Batiuk et al., AST1-3 appear to be mature astrocyte subtypes. AST1 is found in the subpial layer and hippocampus. AST2 has high expression in cortical layers 2/3 and 5, lower expression in cortical layers 1, 4, and 6, and negligible hippocampal expression. AST3 subtype is expressed throughout cortical layers and hippocampus. AST4 is found predominately in the subgranular zone of the hippocampus and may represent a progenitor or hippocampal neural stem cell population [76]. AST5 subtype expression overlaps substantially with AST4 and is hypothesized to represent an intermediate state between progenitor and mature astrocyte. AST5 shows enrichment in cortical layer 2/3 and 5, in the stratum lacunosum-moleculare and dentate gyrus of the hippocampus, and subpial layers.

### 2.2. Mouse Disease Astrocyte Gene-Sets

Astrocytes gene-sets (GSE 35338) from a mouse model of stroke (middle cerebral arterial occlusion (MCAO), 1 day timepoint) and a mouse model of neuroinflammation (lipopolysaccharide (LPS) 1 day timepoint) as described in [43], were obtained. Gene-sets for mouse disease model astrocytes are listed in Appendix A.

### 2.3. Human Astrocyte Gene-Sets

FPKM gene expression values were obtained from BrainRNAseq (www.brainrnaseq.org, access on 3 October 2022) from enriched human mature astrocytes and whole cortex homogenate samples [67]. Genes with mean FPKM less than 1 across all samples were removed. To identify genes enriched in human astrocytes, the average log fold change (LFC) in gene expression in mature astrocyte samples relative to the total homogenate was calculated. Genes were binned into six different groups based on difference in expression. AST_Bin 1: LFC 0-1, AST_Bin 2: LFC 1-1.5, AST_Bin 3: LFC 1.5-2, AST_Bin 4: LFC 2-2.5, AST_Bin 5: LFC 2.5-3, AST_Bin 6: LFC 3-10. Gene-sets for human astrocytes are listed in Appendix A.

### 2.4. Schizophrenia hiPSC-Derived Astrocytes

Three gene-sets of human induced pluripotent stem cells (hiPSC) derived into astrocytes were obtained from [77]. The first gene set consists of 1720 differentially expressed genes (adj *p* < 0.1) from subtype (SUB) compared to classical (CLA) hiPSC astrocytes (AST_hiPSC_SUBvCLA). Subtypes astrocytes are defined as GFAP+ and S100B+ cells with a distinct, non-bushy morphology. The second gene-set consists of 504 differentially expressed genes (adj *p* < 0.1) from schizophrenia (SCZ) compared to healthy control (HC) hiPSC astrocytes (AST_hiPSC_SCZvHC). The third gene-set consists of 430 differentially expressed genes (*p* < 0.05) from schizophrenia clozapine responders compared to clozapine non-responders (AST_hiPSC_SCZ-RvSCZ-NR). Gene-sets are listed in Appendix A.

### 2.5. Disease-Disease Similarity

Curated disease-associated gene-sets for major psychiatric disorders major depression (MDD), bipolar disorder (BPD), and SCZ were downloaded from DisGeNET [78]. To avoid size related bias and improve the specificity of pathway profiles [79,80,81], we further restricted our analysis to diseases with gene-set size between 10 to 500 significant differentially expressed genes. Pairwise-overlap between disease-associated gene-sets was calculated using hypergeometric [82] Gene-Overlap R package version 1.26.0 (http://shenlab-sinai.github.io/shenlab-sinai/, accessed on 14 September 2022). Disease-specific curated gene-lists were downloaded from DisGenNET (https://www.disgenet.org/downloads, accessed on 14 September 2022), and see Appendix A.

### 2.6. Gene Ontology (GO) Analysis

Pathways affected in different astrocyte subtypes and diseases were determined using hypergeometric overlap analysis (HGA) with a background of 21,196 genes (default, GeneOverlap package). The astrocyte-associated gene-sets and disease-associated gene-sets were tested against GO pathways associated with Biological Process (GOBP), Molecular Function (GOMF), and Cellular Component (GOCC). Updated lists of GO-pathways were obtained from the Bader lab (http://download.baderlab.org/EM_Genesets/, accessed on 16 April 2021). To compare enrichment of pathways across different astrocyte subtypes and diseases, the -log10(*p*-value) was used to generate the heatmap. To better identify the character of biological changes in the overlap results, a focused analysis of forty a priori functional themes was performed. As described in our previous study [83], the pathways were filtered based on the parent–child association between GO-terms in our list of significant pathways (child-pathways) and hand-picked parent-pathways representing the a priori theme from the GO-database. Gene-sets for GO analysis are listed in Appendix A.

### 2.7. Density-Index

To quantitatively summarize how common (close to 1) or unique (close to 0) a theme is across different astrocyte subtypes and diseases, we deployed a previously developed density-index [84]. For a given *r* × *c* matrix of -log10(*p*-value), a density-score is obtained as:$$Density=1-\left(\frac{count\;of\;\;zero\;elements\;in\;the\;matrix}{r\times c}\right)$$
where *r* and *c* represent the number of pathways in a theme and number of astrocyte subtypes and diseases, respectively. A zero element represents a non-significant disease-pathway relationship. For density associated with individual pathways and drug signatures, the density-index representing the fraction of non-zero elements in the number of astrocyte subtypes and diseases was used. In such an instance, the numerator of the above density formula represents count of zero elements in a vector. Whereas *r* and *c* in the denominator are constant holding a fixed value of 1 and 27, respectively.

### 2.8. Drug-Target Enrichment Analysis

Enrichment of astrocyte subtypes and drug-induced molecular signatures in the disease-associated gene-sets was calculated using HGA. Drug-specific gene markers were downloaded from the DSigDB library of gene-sets [85]. In order to understand the druggable-mechanism and targets involved, gene-markers of drugs with known modes of action (MOA) and targets were used. Drugs were then further sorted by the typical diseases which the drug is used to treat. DSigDB gene-sets are listed in Appendix A.

### 2.9. Rat Studies

All animal studies were carried out in accordance with the IACUC guidelines at the University of Alabama at Birmingham. Male Sprague Dawley rats were housed with a 12-h cycle of light and darkness. They were pair-housed and given access to food and water ad libitum. Rats were administered with either 28.5 mg kg^−1^ haloperidol-decanoate (*n* = 10) via intramuscular injection every 3 weeks for 9 months or vehicle-treated with sesame oil (*n* = 10) under the same conditions. At the end of the 9 months, brains were removed and stored at −80 °C.

### 2.10. qPCR Studies

RNA was extracted (RNeasy Mini Kit, Qiagen, Germantown, MD, USA) from 14 µm thick fresh frozen sections of rat frontal cortex as previously described [86,87]. RNA was reverse-transcribed (High-Capacity cDNA Reverse Transcription Kit, Applied Biosystems, Life Technologies, Carlsbad, CA, USA) at cycling conditions: 1 cycle: 25 °C for 10 min; 2 cycles: 37 °C for 60 min; 1 cycle: 85 °C for 5 min. cDNA was diluted and equalized to 11 ng/ul. Three microliters of cDNA and 17 µL of Taqman master mix (Applied Biosystems, Life Technologies, Carlsbad, CA, USA) containing 1 µL of Taqman primer were loaded per well and ran at the following cycling conditions: 1 cycle: 95 °C for 10 min; 40 cycles: 95 °C for 15 s, 60 °C for 60 s. Taqman primers for rats assayed were: GFAP (Rn01253033_m1), VIM (Rn00667825_m1), and SOX9 (Rn01751070_m1). The samples were run in duplicate. The relative gene expression was calculated by comparing to a standard curve made of a pool of cDNA from all samples. Gene expression was then normalized to the geomean of reference genes PPIA (Rn00690933_m1) and B2M (Rn00560865_m1). Glial fibrillary acidic protein (GFAP), vimentin (VIM) and Sex-determining region Y (SRY)-box 9 (SOX9) are commonly used astrocyte markers. GFAP is highly expressed in many reactive astrocytes and is upregulated in response to neurological disorders [45]. VIM is a cytoskeletal astrocyte marker and is upregulated in response to neurological injury and is also expressed by progenitor astrocytes [45]. SOX9 is a transcription factor whose expression also increases in response to disease or injury to astrocytes. [45]. Data were analyzed for normal distribution (D’Agostino and Pearson test) and equality of variance (F-test). Data were log transformed and analyzed by Student’s *t*-test. α = 0.05 for statistical tests. Data were analyzed with GraphPad Prism v7.04 (GraphPad Software, San Diego, CA, USA).

### 2.11. “Look up” Studies

The R shiny application “Kaleidoscope” (https://kalganem.shinyapps.io/BrainDatabases/, accessed on 1 June 2022) [88], was used to search astrocyte subtype gene-sets (Appendix A), including expression of GFAP, VIM, SOX9 and excitatory amino acid transporter 2 (EAAT2, gene name SLC1A2) (Appendix A) in publicly available transcriptomic datasets. The data is reported from analysis of rodent brain tissue following at least 2 weeks administration with typical or atypical antipsychotics, antidepressants or mood stabilizers. To determine the effects of antipsychotic medications on astrocyte marker gene expression in human brain, gene names GFAP, VIM, SOX9 and SLC1A2 (EAAT2) were searched in the Stanley Medical Research Institute (SMRI) Online Genomics Database repository [89,90]. The SMRI contains meta-analysis of postmortem brain transcriptomic data from different brain regions (BA6, 8/9, 10, 46, cerebellum) from patients diagnosed with schizophrenia and bipolar disorder who were either on or off antipsychotics and mood stabilizers, respectively, at time of death. The fold change and *p*-value of transcript expression in patients on/off medication is reported. No data is available from the SMRI for MDD patients on or off antidepressant medications.

### 2.12. Identifying Small Molecules to Reverse Astrocyte-Disease Gene Signatures

The DrugFindR R Shiny application was run as described [91] to identify chemical perturbagens that are highly discordant to the AST_hiPSC_SCZvHC dataset. DrugFindR mines the Library of Network Integrated Signatures (iLINCS) chemical perturbagen repository. Antipsychotic drugs enriched in MCF7 or PC3 cell lines in gene enrichment analysis were not included if identified by DrugFindR analysis to prevent redundancy of findings, as DSigDB also utilizes a subset of LINCS signatures [85]. The reported Discordance Score is the negative Pearson correlation coefficient between the disease (AST_hiPSC_SCZvHC) signature and the precomputed iLINCS drug signatures. iLINCS identifies candidate drugs with a score range >+0.2 and <−0.2. We present only those drugs identified with a negative (discordance) score. The lower the discordance score, the more effectively a drug reverses gene expression changes associated with the AST_hiPSC_SCZvHC signature.

### 2.13. Protein kinase Activity Array

Brain tissue from a non-psychiatrically ill control subject (male, age 73, pH 6.4, PMI 17 h, dorsolateral prefrontal cortex) was obtained from the Alabama Brain Collection with consent from next of kin with IRB approval protocols. Approximately 500 mg of brain tissue was homogenized in 5 mL buffer (0.32 M sucrose and 1 mM EDTA, pH 7.4, + HALT Protease Inhibitor) and centrifuged at 1000× *g* for 10 min at 4 °C. The supernatant was layered onto 7.8 mL each of 2%, 6%, 10%, and 20% *v*/*v* Percoll in SEDH buffer (0.32 M sucrose, 1 mM EDTA, 0.25 mM DTT, and 20 mM HEPES, pH 7.4) and centrifuged at 33,500× *g* for 5 min at 4 °C. The turbid layer between the 2% and 6% Percoll layers was collected, resuspended in SEDH buffer, and centrifuged at 1000× *g* for 20 min at 4 °C. The supernatant-containing gliosome fraction was centrifuged at 33,500× *g* for 40 min at 4 °C. The pellet was resuspended and washed ×2 using the same spin conditions. The pellet was resuspended in MPERS buffer. Two micrograms protein from the enriched gliosome fraction and total homogenate was loaded on a Pamgene chip and analyzed using the Pamgene12 (Pamgene International, ‘s-Hertogenbosch, The Netherlands) as previously described [92,93,94]. 

## 3. Results

### 3.1. Astrocyte Subtype Enrichment in Psychiatric Disorders

To determine whether different astrocyte subtypes are enriched in psychiatric disorders, we assessed the enrichment of (1) genes representing six mouse astrocyte subtypes (AST1-5 and “common”) [76], (2) genes representing astrocytes from two mouse disease models (AST_MCAO and AST_LPS) [43], (3) genes (in 6 bins) representing human astrocytes (AST_Bin 1-6) [67] and (4) genes representing iPSC-derived astrocytes from classical and subtype astrocytes (AST_hiPSC_SUBvCLA), astrocytes from SCZ and HC subjects (AST_hiPSC_SCZvsHC) and astrocytes from SCZ subjects who were clozapine responders and non-responders (AST_hiPSC_SCZ-RvsSCZ-NR) [77]. Astrocyte gene-set enrichment was examined in the DisGeNET gene-sets representing different psychiatric disorders (Figure 1).

The set of genes representing the “common” (see details in 2. Materials and Methods) mouse astrocyte subtype were the most frequently enriched, in line with findings that astrocytes play important roles in the pathophysiology of these disorders [66]. The AST4 and AST5 subtypes, proposed to represent progenitor astrocytes and intermediate (progenitor-mature) astrocyte populations, respectively [76], were enriched in a subset of depression and schizophrenia datasets. This may suggest that specific astrocytes are altered in particular disease conditions. The human astrocyte genes contained in bin 6 (AST_Bin 6), which consists of genes with the highest log fold change (LFC 3-10) in expression in human astrocytes relative to total brain, also showed enrichment in schizophrenia but not across the depression datasets. This may be a limitation of the specificity of the “proxy” human astrocyte dataset, as common enrichment was only found with the disease dataset with the highest density index, schizophrenia. Conversely, enrichment of the astrocyte genes associated with the mouse MCAO and LPS disease models and AST_hiPSC genes, in schizophrenia and depression datasets may indicate expression of a common subset of disease–related astrocyte genes whose expression is induced in pathological conditions. Overall, schizophrenia is the disorder with the highest density index (>0.5). As the density index quantitatively compares the significance of disease gene expression to an independent variable (astrocyte population) [84], this suggests greater dysregulation of astrocytes in this disorder compared to other psychiatric disorders, with distinct differences in gene enrichment for certain astrocyte subtypes and in astrocytes in pathological conditions.

### 3.2. Biological Pathway Enrichment across Astrocyte Subtypes and Neuropsychiatric Disease

Biological pathway enrichment (GOBP, GOMF and GOCC) of astrocyte datasets and psychiatric disorders is shown in Figure 2. Parent–child analysis was applied to cluster pathways [83]. There is enrichment of “immune” related pathways in the astrocyte gene-sets from mouse disease models, SCZ AST_hiPSC (response to cytokine pathways) and psychiatric disorders, but limited enrichment of mouse astrocyte subtypes or human astrocytes, supporting unique changes in astrocyte gene expression associated with pathological conditions. In the metabolism cluster, “carbohydrate metabolism” is enriched in astrocyte subtypes and psychiatric disorders but has limited enrichment in mouse disease models or human hiPSC astrocytes datasets. This may suggest that pathological changes in carbohydrate metabolism are not driven by astrocytes in disease states. “Peptide metabolic process” is highly enriched in AST4 but not other astrocyte subtypes and “response to oxidative stress” is enriched in AST4 and AST5 subtypes and MCAO model but not other subtypes. This supports the idea that different astrocyte subtypes may be implicated in specific biological processes. Conversely, lipid metabolism appears to be an essential function of astrocytes, highly enriched in both the mouse “common” astrocyte subtype and the human AST_Bin 6 dataset. “Proteolyis” pathways are enriched across astrocyte and psychiatric disorder datasets but not in AST_hiPSC datasets which suggests that proteolysis is not perturbed in astrocytes in schizophrenia, or alternatively, may result from differences in gene expression in hiPSC-derived astrocytes compared to astrocytes from in vivo models and human brain tissue.

Signaling cluster pathways, including “modulation of chemical synaptic transmission”, “regulation of membrane potential” and “regulation of neurotransmitter levels” are commonly enriched in AST_hiPSC datasets and across psychiatric disorder datasets but have limited enrichment in mouse subtype or human astrocyte datasets, suggesting species and disease specific alteration in astrocyte-related signaling processes. High levels of gene enrichment in the “Synapse” subcluster were largely driven by the hiPSC- derived astrocyte gene-sets and support changes in synaptic transmission and astrocyte sensitivity to synaptic signaling in different human astrocyte subtypes (hIPSC_SUBvCLA), in disease (hIPSC_SCZvHC) and in response to medication (hIPSC_SCZ-NRvSCZ-R) [77]. These findings also implicate altered glutamate system signaling, of which astrocytes are essential regulators via glutamate synthesis and clearance mechanisms [77,95]. “G protein coupled receptor (GPCR) signaling pathways” are highly enriched in psychiatric disorder datasets but have limited enrichment across astrocyte-related datasets. Although astrocytes express GPCRs [96], these results suggest that deficits in GPCR signaling in psychiatric disorders are not related to their astrocyte function. Conversely, “regulation of protein phosphorylation” pathways are enriched across datasets, suggesting a fundamental role for cellular regulatory processes in astrocytes and psychiatric disorders. Overall, pathway analysis identified biological processes that are commonly dysregulated in psychiatric disorders [53,54,55,56,57,58,59], and supports a role for astrocytes in a subset of these pathological processes. 

### 3.3. Psychotropic Medication Effect on Astrocyte Subtypes and Disease

Next, we assessed the gene enrichment of astrocyte subtypes, psychiatric disorders, and the gene-sets representing different types of commonly prescribed psychotropic medications (antipsychotics, antidepressants and mood stabilizers) from the DSigDB library (Figure 3). Overall, gene enrichment of the astrocyte datasets and psychotropic drug datasets is low. In contrast, there are high levels of gene enrichment across psychotropic drug datasets and most psychiatric disorder datasets. All medication classes included at least one medication with a relatively high density index (>0.5). This suggests that psychotropic drugs induce significant changes in gene expression that are common to gene expression associated with psychiatric disorder datasets, but not astrocyte datasets. These findings may also indicate an effect of specific medications on specific astrocyte subtypes. For example, gene expression changes associated with the mood stabilizer lithium chloride are enriched in the mouse AST4 subtype, but not other mouse astrocyte subtypes. Equally, the AST_hiPSC_SUBvsCLA dataset shows low levels of gene enrichment with antidepressant and antipsychotic datasets but moderate levels of gene enrichment with mood stabilizer datasets. These results may suggest that this class of drugs have unique effects on this particular astrocyte subtype. Although drug gene-set enrichment across schizophrenia disease datasets is lower than other disease datasets, there is still common enrichment with specific astrocyte gene-sets and drug datasets (e.g., haloperidol_BOSS, thioridazine_PC3_Up, clozapine_CTD), suggesting that different drugs may target specific subsets of astrocytes and may be a promising therapeutic strategy in this disorder.

The DSigDB is a collection of drug datasets from different sources and include data generated from varied cell types and tissues that use a range of drug concentrations and dosage time points that may contribute to heterogeneous results. This is reflected in the gene enrichment heatmaps presented here, where common enrichment of astrocyte is highly variable across different drug datasets but is relatively consistent across disease datasets, suggesting robust gene enrichment in disease but not astrocytes.

### 3.4. Effect of Chronic Medication on Astrocyte Marker Expression in Rat Brain

To further explore the relationship between psychotropic medications and their effects on astrocytes, we carried out a “look-up” study of antipsychotics, antidepressants and mood stabilizers on an extensive range of astrocyte marker genes in rodent frontal cortex. We used those genes defined in the “mouse_common” gene list, which is comprised of genes found in at least 60% of astrocytes analyzed by [76]. Figure 4A and Appendix A shows gene expression (Log2FC) of these astrocyte marker genes in rodent brain following chronic (2 weeks–12 weeks) administration with mood stabilizers, antidepressants or antipsychotics. The largest changes in astrocyte marker expression are induced by mood stabilizers (lithium and valproate) and the weakest effect is seen in antidepressant datasets. Interestingly, gene expression changes following 4 weeks or 12 weeks of antipsychotic (haloperidol and clozapine) administration differ significantly, as previously observed [97]. To further explore differences in the chronic effects of antipsychotics on astrocyte-associated gene expression, we assayed the mRNA levels of three established astrocyte markers, GFAP, VIM, and SOX9, in the frontal cortex of rats administered the typical antipsychotic haloperidol-decanoate for 9 months (36 weeks), as this time-point better models a chronic treatment course in schizophrenia patients. As at the 4- and 12- week time-points (Appendix A), no significant differences were detected in the mRNA expression of GFAP, SOX9, or VIM in haloperidol-decanoate treated rats compared to vehicle-treated controls (*p* > 0.05; Figure 4B–D). We have previously reported no significant change in the mRNA expression of the rodent EAAT2 homolog GLT-1, another common astrocyte marker, in the frontal cortex of rats treated chronically with haloperidol-decanoate [86]. These results suggest that chronic haloperidol administration does not significantly alter astrocyte gene marker expression, i.e., it does not make them more reactive. It also highlights the importance of studying the effects of chronic medication treatment in a manner that better recapitulates patient treatment courses. Conversely, GFAP, VIM, and SOX9 mRNA expression were significantly increased in schizophrenia subjects who were “on” antipsychotic medication at time of death (data obtained from SMRI database) but there was no significant difference in astrocyte marker transcript expression in bipolar disorder subjects who were “on” mood stabilizers compared to subjects who were “off” (Figure 4E). These results highlight the importance of conducting translational studies to elucidate the effects of chronic psychotropic medication on the brain of psychiatric patients, as well as studying the effects of chronic medication treatment in a manner that better recapitulates patient treatment courses.

### 3.5. Exploratory Studies of Common Protein Kinase Activity Profiles in Postmortem Glial Cell Compartments and in Schizophrenia

Protein kinase activity, determined by an increase in phosphorylation signal intensity, is higher in a gliosome preparation from postmortem frontal cortex tissue relative to total homogenate from the same sample (Figure 5A,B). This is in line with pathway analysis, which suggests that protein phosphorylation activity is significantly enriched across astrocyte populations (see arrowhead, Figure 2). A subset of the kinases identified in the gliosome preparation (Figure 5C) were also identified in our earlier study of kinase activity in schizophrenia postmortem frontal cortex tissue [94], suggesting that changes in astrocyte kinase activity may contribute to signal transduction network perturbations in this disorder. The gliosome preparation represents kinase activity in a single astrocyte compartment and does not exclude a role for dysregulated kinase activity from other astrocyte subcellular compartments in schizophrenia.

### 3.6. Chemical Perturbagens That Reverse the hiPSC_SCZvHC Gene Signature

Using DrugFindR, we identified the FDA approved drugs with the greatest discordance (≤−0.60) with the hiPSC_SCZvHC gene signature (Table 1). This human astrocyte dataset was selected as the best gene-set to identify human disease-relevant discordant chemical perturbagens. Several antipsychotics were identified, although only one, promazine, had a discordance score <−0.60, suggesting that while gene signatures induced by antipsychotics are negatively correlated with the SCZ_AST_hiPSC gene signature, these effects are likely modest, in line with the relatively limited evidence of gene enrichment of astrocyte and psychotropic drug datasets in Figure 3. Interestingly, amongst the other classes of drugs identified in the DrugFindR analysis, several kinase inhibitors (<−0.6) were identified. Pathway analysis suggests that protein phosphorylation activity is dysregulated across psychiatric disorder datasets and different astrocyte populations (Figure 2), and these findings implicate a therapeutic role for modulation of kinases in schizophrenia [94,98].

## 4. Discussion

Astrocytes play an important, albeit still poorly understood role in many neurological and psychiatric diseases [1,2,3,66]. Although postmortem studies generally support altered astrocyte expression and morphology in these disorders [6,7,10,20,21,22,23,24,25,26,27,28,29], many studies have shown conflicting findings [4,5,6,7,8,9,10,11,12,13,14,15,16,17,18,19] or report no changes in astrocytes in disease [27,30,31,32,33,34,35]. The diversity of astrocyte subtypes [43,46], their responses to different insults [43,49,62], and the potential effects of psychotropic medications on astrocytes [12,15,23,24,25,69,70,71,72,73,74,75] may contribute to challenges in defining and interpreting astrocyte changes in the brain in these disorders.

To improve our understanding of how astrocytes are altered in the brain in disease, and the effects of medications on astrocytes, we carried out bioinformatic analysis, taking advantage of emerging single-cell RNAseq datasets that define mouse astrocyte subtypes. As unique molecular signatures for different subtypes of human astrocytes have yet to be elucidated [66,67], we instead used signatures for five different astrocyte subtypes (AST1-5) and a “common” astrocyte signature derived from mouse brain [76]. A proxy human astrocyte gene-set, derived from the Brain-Seq project [67], was included to account for species differences in astrocytes. Finally, to account for changes in astrocyte gene expression due to inherent differences in astrocytes in “healthy” and “pathological” conditions, we also included astrocyte gene-sets derived from two different mouse models of disease (MCAO model and LPS-induced infection model) [43] and from hiPSC derived astrocytes from schizophrenia patients and healthy controls (astrocyte types, disease differences, clozapine responders and non-responders) [77].

### 4.1. Species, Astrocyte Subtype and Disease-Driven Changes in Gene Enrichment in Psychiatric Disorders

Gene enrichment analysis [84] found that genes associated with different astrocyte datasets were enriched in schizophrenia and depression, supporting a role for astrocytes in these psychiatric disorders. Although the highest level of enrichment across psychiatric disorders was seen with the “common” mouse astrocyte subtype, the AST4-5 subtypes, which are highly expressed in hippocampus and cortical layers II/III and V, and likely represent progenitor or immature astrocytes [76], were also enriched in schizophrenia and depression datasets. Conversely, the human astrocyte gene-sets (AST_Bin 6) were not enriched in depression datasets, but astrocyte gene-sets from the mouse disease models and schizophrenia hiPSC astrocytes were enriched across depression datasets. As changes in astrocyte expression and morphology have been reported in postmortem brain in MDD, including in the hippocampus [11,12,17,19,99,100,101], these findings highlight the species-specific differences in molecular signatures of astrocyte subtypes [102], and the importance of studying astrocytes under pathological conditions to understand the unique changes in this cell type in different disease states [43,49,62].

There is extensive conservation of gene expression between human and mouse astrocytes [102]. However, whereas genes involved in metabolism are highly enriched in mouse astrocytes, genes with higher enrichment in human astrocytes are implicated in processes described by GO term “defense response”, which includes inflammatory responses. Human astrocytes are also more vulnerable to oxidative stress, indicating further differences in mitochondrial metabolism in human and mouse astrocytes [102]. These distinctive physiological responses may contribute to the dissimilarities found in astrocyte gene enrichment in human psychiatric disorders and mouse astrocyte subtypes. Human astrocytes also carry out a wider range of functions including roles in augmenting cognition [103]. Thus, rodent models cannot fully capture the structural, functional and molecular diversity of human astrocyte subtypes, several of which are not found in the rodent brain [66]. Comprehensive molecular profiling of these different, potentially highly specialized cell subtypes in humans, are essential to understand their role in pathology and potential as therapeutic targets.

This is further supported by the diverse patterns of gene enrichment of the mouse disease model astrocyte datasets compared to the astrocyte subtypes from healthy mice. However, differences in astrocyte gene-set enrichment in different disease datasets may also indicate that specific astrocyte subtypes are affected in particular disorders, as seen in mental depression, where mouse AST1, a subtype found in the hippocampus, was uniquely enriched. Overall, our data support a role for astrocytes in psychiatric disorders, but it will be necessary to characterize and study human astrocyte subtypes and their unique responses to different pathological states to fully understand their role in the brain and disease.

Next, we looked at how astrocyte function relates to psychiatric disorders and saw that many of the same biological pathways were enriched in mouse astrocyte subtypes, in the mouse disease model astrocytes, the human astrocytes, and in psychiatric disorders. Pathway analysis identified common enrichment in the “metabolic”, “plasticity/structure” and “signaling” pathway clusters, highlighting the role of astrocytes in these processes and the dysregulation of these processes in schizophrenia and depression [66,104,105]. Unsurprisingly, the “immune” related pathways were highly enriched in both mouse disease model astrocyte datasets compared to the other astrocyte subtypes, and in depression and schizophrenia. A subset of pathways, “response to cytokine”, were also enriched in AST_hiPSC datasets suggesting a targeted immune response in astrocytes derived from schizophrenia patients. Neuroimmune dysregulation is commonly reported in psychiatric disorders and may contribute to the underlying pathophysiology [13,14,66,104,105]. This finding also highlights the functional differences in astrocytes in disease states compared to healthy conditions.

The “carbohydrate metabolism” pathway cluster is enriched in astrocyte subtypes and psychiatric disorders but has limited enrichment in mouse disease models or AST_hiPSC schizophrenia datasets. Astrocytes play an essential role in ensuring the bioenergetic requirements of neurons are met in the brain [106]. We have previously shown significant deficits in the expression and enzyme activity of hexokinase, a rate-limiting enzyme in glucose metabolism, in pyramidal neurons but not in astrocytes, in the brain in schizophrenia [107]. Thus, this gene enrichment analysis confirms a role for astrocytes in carbohydrate metabolism, but does not support major deficits in these biological processes in astrocytes in disease.

Additionally, “regulation of protein phosphorylation” was enriched across astrocyte and psychiatric disorder datasets, implicating these processes in astrocytes under physiological and pathological conditions. We confirmed enrichment of protein kinase activity in a gliosome preparation relative to total homogenate in human postmortem brain tissue, and found that changes in activity in a subset of these kinases are also implicated in schizophrenia [94,98]. This suggests that the gene enrichment associated with astrocytes and psychiatric disorder datasets generated by bioinformatic analysis reflect biological processes associated with these cells and lends additional support for dysregulation of these processes in disease. 

A potentially confounding variable in understanding the role of astrocytes in disease is the effect of psychotropic medications that may be normalizing disease-related changes in astrocytes, or inducing structural, molecular, or functional changes in these cells [23,24,26,73,75,108,109]. Postmortem studies that have considered the effects of psychotropic medications have found changes in astrocyte cell counts [73], morphology [75], and gene expression markers [15]. However, it is not clear if these changes reflect a therapeutic response to medication, an effect of the disease process, or a combination of both. Additionally, if astrocytes are altered in the disease state or as a result of medication, this may support their role as a therapeutic target. To address these questions, we examined enrichment of gene signatures induced by commonly prescribed psychotropic drugs in the different astrocyte and psychiatric disorder datasets. Overall, psychotropic drug gene lists were highly enriched in depression (adding support for off-label use of antipsychotics for the treatment of MDD [110]) and to a lesser extent in schizophrenia, but there was surprisingly limited enrichment across astrocyte datasets. For example, astrocytes express the dopamine D2 receptor [111], the major pharmacological target of currently available antipsychotic medications, and 5-HT2B, a serotonin receptor which is a target of all selective serotonin reuptake inhibitor (SSRI) administration [112,113]. However, our findings suggest that psychotropic drugs do not induce significant changes in astrocyte-associated gene-sets. Higher levels of gene enrichment of the mouse disease model astrocytes and the human astrocyte datasets support the idea that there are inherent differences in the way that psychotropic medications affect astrocytes in different species and in the healthy versus disease state, as discussed [43,49,62]. Notably, we found enrichment of the AST_hiPSC_SCZvsHC gene-set with clozapine drug datasets but not with the AST_hiPSC_SCZ-RvsSCZ-NR (clozapine responder’s vs. non-responders) dataset. This may reflect a limitation of the drug datasets available in DSigDB, or that the threshold for identifying significant differentially expressed genes was lower for this dataset.

While the general lack of consistent enrichment of drug-induced genes in astrocyte datasets is somewhat surprising [114,115], our data also suggest that specific astrocyte subtypes may be affected by specific medications, as seen with the unique enrichment of lithium chloride and the mouse AST4 subtype gene-set and the hiPSC_SUBvsCLA dataset. We also found enrichment of mouse MCAO astrocytes, and lithium chloride gene-sets, in line with a recent report showing a neuroprotective effect of lithium in ischemic stroke [116]. This further supports that different astrocytes have diverse responses to different drugs and pathological conditions.

Overall, these bioinformatic findings suggest that astrocytes are not significant primary targets of psychotropic medications used in the treatment of psychiatric disorders but different subtypes of astrocyte may be targets under specific conditions. In support of this, we found that chronic antipsychotic administration did not significantly affect astrocyte marker (GFAP, VIM, SOX9 and EAAT2) transcript expression in the frontal cortex in rodent brain. Future studies will be required to examine marker expression in other brain regions. “Look-up” studies of rodent transcriptomic brain datasets also found few reports of significantly altered astrocyte marker transcript expression changes in response to different psychotropic medications, suggesting that chronic medication administration has a limited effect on transcript expression of established glial markers in rodent models. Importantly, “look-up” studies did find significant increases in expression of astrocyte gene markers in postmortem brain of schizophrenia subjects who were “on” antipsychotics compared to those who were “off” antipsychotics but not in bipolar disorder subjects who were “on” mood stabilizers compared to those who were “off,” highlighting the importance of considering specific disease- and drug-interaction effects on astrocytes. Future studies should consider a broader range of astrocyte genes to better capture understand the diversity of astrocyte subtypes and their responses in pathological states.

Finally, we mined the LINCS chemical perturbagen gene signature repository to identify FDA-approved small molecules that could potentially reverse the gene expression changes associated with astrocytes in schizophrenia (AST_hiPSC_SCZvHC). Kinase inhibitors were among the top drug hits identified by this analysis and typically had stronger discordance scores than antipsychotics, suggesting that modulating astrocytic perturbation of kinases may have therapeutic potential in schizophrenia. 

### 4.2. Limitations

A necessary limitation of this study is the use of astrocyte subtype molecular signatures derived from healthy mice for bioinformatic analysis, which may not reflect the subtypes of astrocytes found in human brain [76,102]. Additionally, the gene-sets (Bins 1-6) used as a proxy for a human astrocyte gene signature are derived from the log fold change of gene expression in enriched astrocyte populations relative to the expression in total brain homogenate, and may not represent true human astrocyte molecular signatures. The analyses described here utilize transcript-level data. Many drugs act at the protein level rather than at the transcript level and their effects on astrocytes may not be fully captured by gene enrichment analysis. Finally, qPCR and lookup studies used GFAP, VIM, SOX9 and EAAT2 transcript expression as astrocyte markers. These are commonly used astrocyte markers with conserved expression across species that play unique roles in the reactive astrogliosis response [117,118,119,120], but they may not represent all astrocyte subtypes or molecular changes associated with disease. Future studies should also consider how other, lesser studied astrocyte markers are altered in disease states and in response to medication. 

## 5. Conclusions and Future Directions

In summary, this study provides a unique analysis of archived transcriptomic data and emerging molecular characterization of astrocyte subtypes to illuminate the relationship between psychotropic medications and astrocytes in the context of psychiatric disorders. We identified gene enrichment of distinct subtypes of astrocytes in psychiatric disorders but surprisingly, minimal enrichment of psychotropic medication-induced gene expression in astrocyte subtypes. This bioinformatic analysis was supported (generally) by “lookup” studies and qPCR in rodents, suggesting that these medications have limited effects on inducing gene expression changes in astrocytes. However, our results highlight the importance of conducting truly translational studies, including postmortem brain studies that account for the effects of medication, due to the differences in astrocyte gene-sets in different species, cell subtypes and disease states. Interestingly, when we identified the biological processes associated with different astrocyte subtypes, regulation of protein phosphorylation was implicated across astrocyte and disease datasets. In combination with our exploratory studies of gliosome protein kinase activity, and in silico analysis of chemical perturbagens that reverse astrocyte-disease related gene signatures, our findings suggest that dysregulated astrocyte kinase activity is potentially a novel therapeutic target in schizophrenia. However, further work is required to elucidate the role of altered regulation of protein phosphorylation in astrocytes in the brain in psychiatric disorders. These findings serve as a starting point to better understand the complex relationship between psychotropic medications and astrocytes in psychiatric disorders, which could have significant consequences on our understanding of the pathophysiology of disease, and potentially offer insight into targeting astrocytes therapeutically.

## Figures and Tables

**Figure 1 cells-11-03315-f001:**
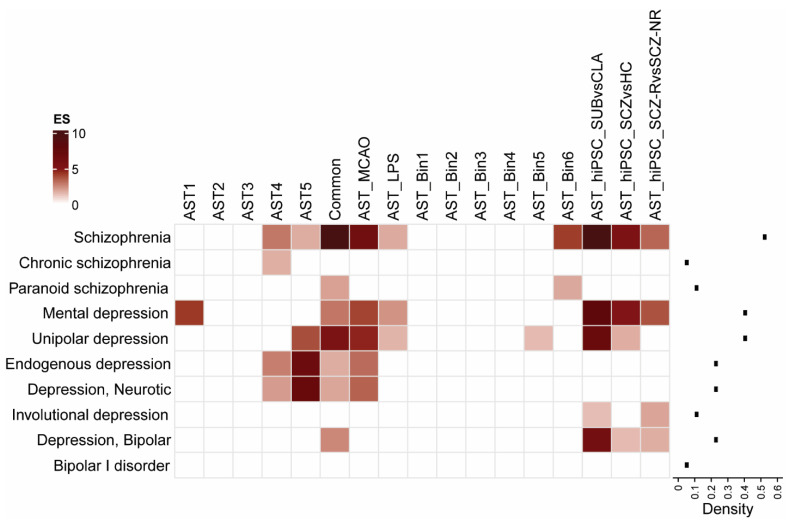
Gene enrichment of mouse astrocyte subtype gene-sets (AST1-5, common), mouse disease model astrocyte gene-sets (AST_MCAO, AST_LPS), and proxy human astrocyte gene-sets (AST_Bin1-6) across neuropsychiatric and neurological disorders (y-axis). Color intensity is proportional to -log10(*p*-value). Density index for all astrocyte gene-sets across all disorders is displayed to right of heatmap. AST astrocyte, CLA astrocyte classical subtype, ES enrichment score, HC healthy control, hiPSC human induced pluripotent stem cell, LPS lipopolysaccharide, MCAO middle cerebral arterial occlusion. SCZ schizophrenia, SCZ_R clozapine responder, SCZ_NR clozapine non-responder, SUB astrocyte subtype.

**Figure 2 cells-11-03315-f002:**
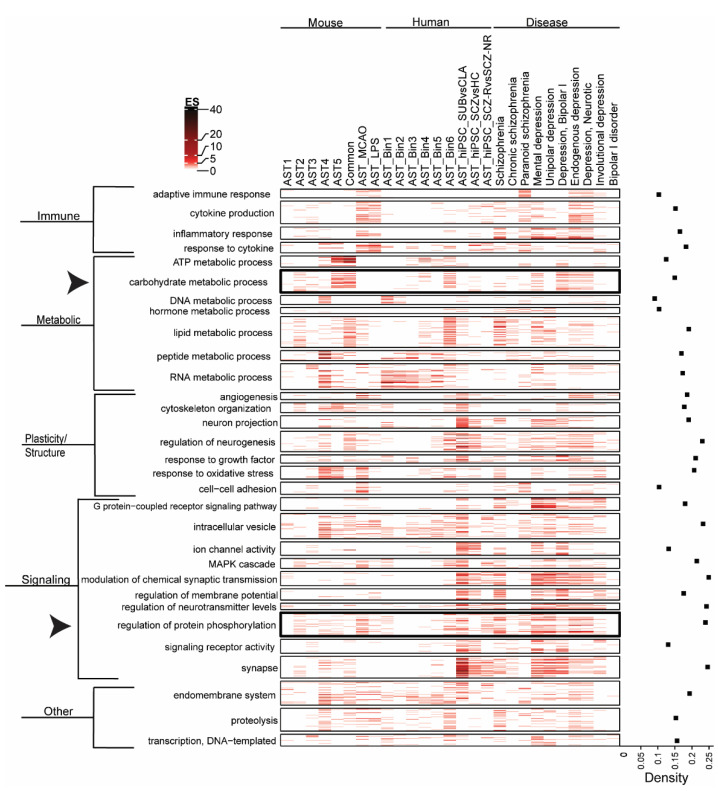
Enrichment of clustered biological pathways across astrocyte gene-sets and neuropsychiatric and neurological disorders. Color intensity is proportional to -log10(*p*-value). Density index for all pathway clusters across astrocyte groups and disorders is displayed to right of heatmap. Arrowhead highlights pathways “regulation of protein phosphorylation” and “carbohydrate metabolism”. AST astrocyte, CLA astrocyte classical subtype, ES enrichment score, HC healthy control, hiPSC human induced pluripotent stem cell, LPS lipopolysaccharide, MCAO middle cerebral arterial occlusion. SCZ schizophrenia, SCZ_R clozapine responder, SCZ_NR clozapine non-responder, SUB astrocyte subtype.

**Figure 3 cells-11-03315-f003:**
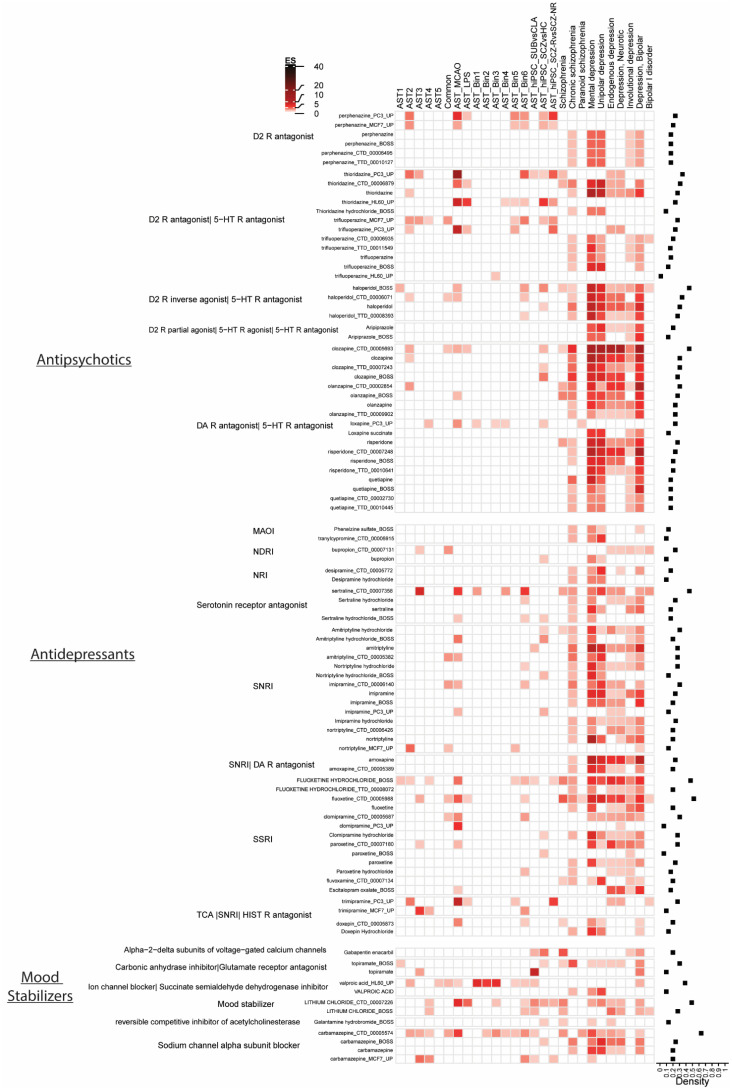
Gene enrichment of different classes of psychotropic medications across astrocyte gene-sets and neuropsychiatric and neurological disorders. Color intensity is proportional to -log10(*p*-value). Density index for all medications across astrocyte groups and disorders is displayed on right. AST astrocyte, CLA astrocyte classical subtype, ES enrichment score, HC healthy control, hiPSC human induced pluripotent stem cell, LPS lipopolysaccharide, MCAO middle cerebral arterial occlusion. SCZ schizophrenia, SCZ_R clozapine responder, SCZ_NR clozapine non-responder, SUB astrocyte subtype.

**Figure 4 cells-11-03315-f004:**
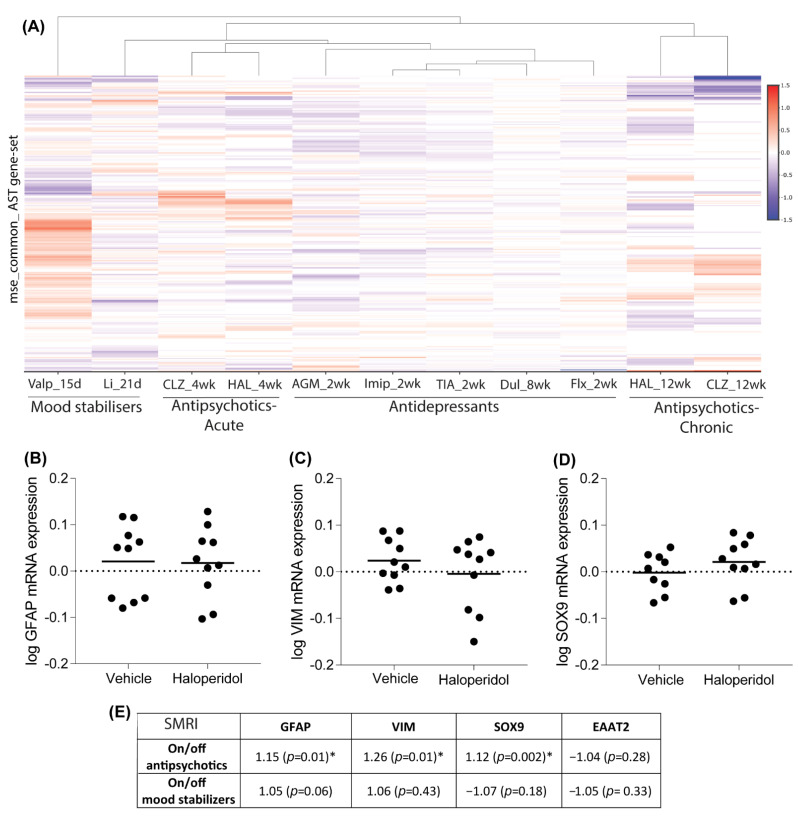
(**A**) Heatmap of log 2 fold change transcript expression of astrocyte marker gene list, derived from mouse_common gene-set, in models of psychotropic drug administration. Differences in patterns of gene expression were identified in 12 weeks compared to 4 weeks of antipsychotic administration. In rats treated for 36 weeks (9 months) with the antipsychotic haloperidol-decanoate (*n* = 10/group), frontal cortex mRNA expression of astrocyte markers (**B**–**D**) GFAP, VIM, and SOX9 was not significantly different (Student’s *t*-test, *p* > 0.05) compared to vehicle treated control (*n* = 10/group). (**E**) Descriptive statistics of gene expression of GFAP, VIM, and SOX9 in the chronic antipsychotic-treated rat study. Data was log-transformed. Data expressed as mean. GFAP glial fibrillary acidic protein, VIM vimentin, SOX9 SRY (sex-determining region Y)-box 9. * *p* < 0.05.

**Figure 5 cells-11-03315-f005:**
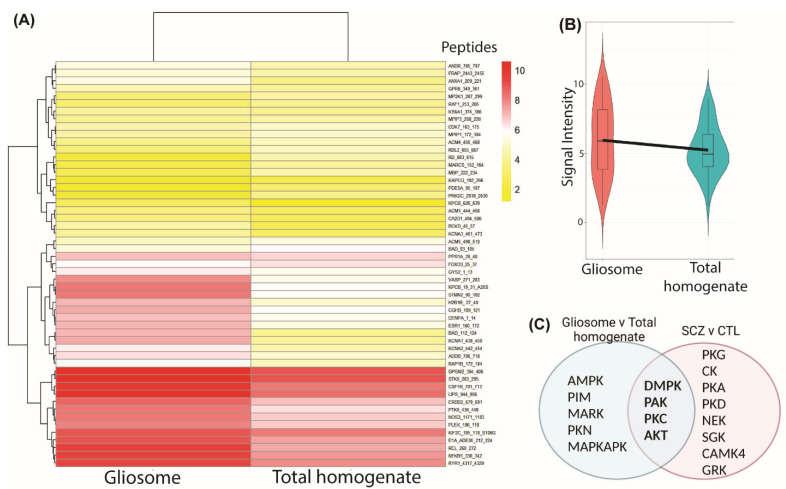
(**A**) Heatmap of Pamgene12 kinome array peptide phosphorylation intensity in a gliosome fraction and total homogenate from postmortem dorsolateral prefrontal cortex. A greater number of peptides are phosphorylated in the gliosome fraction suggesting enrichment of serine/threonine protein kinase activity in this sample. (**B**) Violin plot of peptide phosphorylation signal intensity in gliosome fraction and total homogenate. (**C**) Following peptide mapping, a number of common kinases with differential activity (L2FC +/− 0.2) in the gliosome relative to total homogenate samples and a previous study of postmortem schizophrenia (SCZ) and control (CTL), were identified.

**Table 1 cells-11-03315-t001:** FDA approved drugs that are discordant (discordance score <−0.6) with the AST_hiPSC_SCZvHC gene signature. Drugs were identified using the iLINCS repository of chemical perturbagens. A number of antipsychotics, kinase inhibitors and “other” drugs were identified.

Drug	Score	LINCS ID	Cell Line	Dose	Timepoint	Drug Class
**Antipsychotics**						
Promazine	−0.656	LINCSCP_235798	MCF7	10 uM	6 h	antipsychotic
Chlorpromazine	−0.493	LINCSCP_21936	HEPG2	10 uM	6 h	antipsychotic
Trifluoperazine	−0.474	LINCSCP_64508	VCAP	5 uM	24 h	antipsychotic
Phenothiazine	−0.464	LINCSCP_209948	HA1E	10 uM	24 h	antipsychotic
Fluphenazine	−0.460	LINCSCP_22209	HEPG2	10 uM	6 h	antipsychotic
Thioridazine	−0.511	LINCSCP_50729	RKO	10 uM	6 h	antipsychotic
Trifluoperazine	−0.474	LINCSCP_64508	VCAP	5 uM	24 h	antipsychotic
**Kinase inhibitors**						
Neratinib	−0.673	LINCSCP_163133	SKBR3	10 uM	3 h	kinase inhibitor
Vemurafenib	−0.658	LINCSCP_162826	SKBR3	1.11 uM	24 h	kinase inhibitor
Tivozanib	−0.653	LINCSCP_128031	HUES3	0.37 uM	24 h	kinase inhibitor
Everolimus	−0.618	LINCSCP_121310	HT29	0.37 uM	24 h	kinase inhibitor
Dacomitinib	−0.602	LINCSCP_84161	ASC	3.33 uM	24 h	kinase inhibitor
**Other drugs**						
Oxandrolone	−0.689	LINCSCP_34288	MCF7	10 uM	24 h	Anabolic steroid
Vorinostat	−0.636	LINCSCP_67113	HT29	10 uM	6 h	HDAC inhibitor
Cholic acid	−0.631	LINCSCP_258416	NEU	10 uM	24 h	Bile acid
Aminosalicylic acid	−0.631	LINCSCP_141713	MCF7	10 uM	24 h	Antitubercular agent
Tadalafil	−0.623	LINCSCP_144322	MCF7	0.04 uM	24 h	Phosphodiesterase 5 inhibitor
Omacetaxine mepesuccinate	−0.623	LINCSCP_126799	HT29	0.37 uM	24 h	protein synthesis inhibitor
Abiraterone acetate	−0.613	LINCSCP_153115	PC3	10 uM	24 h	antiandrogen
Mebendazole	−0.611	LINCSCP_27148	HT29	10 uM	6 h	anthelmintic
Bisacodyl	−0.609	LINCSCP_30211	MCF7	10 uM	6 h	stimulant laxative

## Data Availability

Publicly available datasets accessed in this study: DisGenNET (https://www.disgenet.org/downloads); Gene Ontology annotation (http://download.baderlab.org/EM_Genesets/); DrugFindR (https://github.com/CogDisResLab/drugfindR); Kaleidoscope (https://kalganem.shinyapps.io/BrainDatabases/). All other genesets available directly from cited material.

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
