# Peer review of "Gene Enrichment Analysis of Astrocyte Subtypes in Psychiatric Disorders and Psychotropic Medication Datasets"

_cells, 2022, doi:10.3390/cells11203315_

Round 1

Reviewer 1 Report

This is a very interesting study making use of the abundant transcriptomic datasets available. The authors address the question of what is the relationship between the different types of astrocytes with psychiatric disorders and the medications use to treat them.

The study is well designed, the results are congruent with the analysis and the available literature. It also provide new insights into the potential role of medications with astrocytes in psychiatric diseases.

I have only minor comments.

It is important to explicitly define the discordant score and the boundaries used to denote a score as high.

only one subject was used for gliosome study, and itis no clear to this reviewer if the donor has any diagnosis that includes astrocyte reactivity, or if it was considered as a control case.

In figure 3, SZ has the least gene enrichment compared with other disorders however, it is the more concordant with different astrocytes database for particular medications, please discuss.

Synapse receptor activity was module with the highest density index, however no discussion was included in the manuscript.

Author Response

Reviewer 1.

This is a very interesting study making use of the abundant transcriptomic datasets available. The authors address the question of what is the relationship between the different types of astrocytes with psychiatric disorders and the medications use to treat them.

The study is well designed, the results are congruent with the analysis and the available literature. It also provide new insights into the potential role of medications with astrocytes in psychiatric diseases.

I have only minor comments.

 Response. We thank the reviewer for their comments to improve the manuscript.

It is important to explicitly define the discordant score and the boundaries used to denote a score as high.

Response. We thank the reviewer for their comment. This was added to section 2.12, page 6.

Only one subject was used for gliosome study, and it is no clear to this reviewer if the donor has any diagnosis that includes astrocyte reactivity, or if it was considered as a control case.

Response. We thank the reviewer for their comment and confirm (page 6 section 2.13) that this tissue is from a non-psychiatrically ill control subject.

In figure 3, SZ has the least gene enrichment compared with other disorders however, it is the more concordant with different astrocytes database for particular medications, please discuss.

Response. We have added additional discussion of the SCZ enrichment and Fig 3 drug data in section 3.3 page 9-10 and 4.1 page 17, paragraph 1.

Synapse receptor activity was module with the highest density index, however no discussion was included in the manuscript.

Response. We thank the reviewer for their comment and now add discussion on this observation (section 3.2 page 9).

Reviewer 2 Report

The manuscript by Zhang et al uses an interesting combination of bioinformatic analysis of several existing gene expression profiling datasets and confirmation with both lookup studies and qRT PCR to examine gene expression profiling of astrocyte subtypes in subjects with schizophrenia and how much of the observed gene expression changes may be due to exposure to antipsychotics. This is an excellent study elegantly conducted. The data address an important and timely question and the finding that astrocyte subtypes may not be the primary target of antipsychotic medications represents a significant contribution to the field and highlights several key questions going forward, as well as the potential of this approach for future studies. There are a few minor points discussed below that can improve the manuscript. 

1)    In the methods section, under the description for lookup studies, the authors state that they searched the Stanley Medical Research Institute Online Genomics Database repository. Which brain areas does this analysis reflect? The SMRI includes data from several brain regions which can be used to compare region-specific effects, for example between frontal cortex and hippocampus. This may be useful to include in the manuscript.

2)    Rat studies examining effects of psychiatric medications on gene expression were focused on frontal cortex. Are datasets from other brain regions available to test for region specific effects? Along this line, in the discussion the authors state 

“In support of this, we found that chronic antipsychotic administration did not significantly affect astrocyte marker (GFAP, VIM, SOX9 and EAAT2) transcript expression in rodent brain.” 

This statement should be revised to clarify that this was in the frontal cortex, as region specific changes were not ruled out. 

3)    An important limitation that the authors discuss is the focus on using mouse astrocyte subtype molecular signatures from healthy brain tissue due to the lack of availability of equivalent human signatures. The authors clearly state this limitation and highlight the need for this data from human astrocytes. However, additional discussion on potential human specific astrocyte subtype and gene expression differences would be helpful, particularly in light of several recent studies pointing to human specific enhanced complexity of astrocytes. 

Author Response

Reviewer 2.

The manuscript by Zhang et al uses an interesting combination of bioinformatic analysis of several existing gene expression profiling datasets and confirmation with both lookup studies and qRT PCR to examine gene expression profiling of astrocyte subtypes in subjects with schizophrenia and how much of the observed gene expression changes may be due to exposure to antipsychotics. This is an excellent study elegantly conducted. The data address an important and timely question and the finding that astrocyte subtypes may not be the primary target of antipsychotic medications represents a significant contribution to the field and highlights several key questions going forward, as well as the potential of this approach for future studies. There are a few minor points discussed below that can improve the manuscript. 

 Response. We thank the reviewer for their comments to improve the manuscript.

1)    In the methods section, under the description for lookup studies, the authors state that they searched the Stanley Medical Research Institute Online Genomics Database repository. Which brain areas does this analysis reflect? The SMRI includes data from several brain regions which can be used to compare region-specific effects, for example between frontal cortex and hippocampus. This may be useful to include in the manuscript.

Response. We thank the reviewer for their comment. We have added description of the different brain regions included in the meta-analysis data provided in the SMRI for on v off medication SCZ study (section 2.11. “Look-up” studies).

2)    Rat studies examining effects of psychiatric medications on gene expression were focused on frontal cortex. Are datasets from other brain regions available to test for region specific effects? Along this line, in the discussion the authors state 

“In support of this, we found that chronic antipsychotic administration did not significantly affect astrocyte marker (GFAP, VIM, SOX9 and EAAT2) transcript expression in rodent brain.” 

This statement should be revised to clarify that this was in the frontal cortex, as region specific changes were not ruled out. 

Response. As suggested, we clarify that these results were specific to the frontal cortex in rodent brain and that additional assays are required for marker expression in other brain regions (Discussion section 4.1, page 17, paragraph 3).

3)    An important limitation that the authors discuss is the focus on using mouse astrocyte subtype molecular signatures from healthy brain tissue due to the lack of availability of equivalent human signatures. The authors clearly state this limitation and highlight the need for this data from human astrocytes. However, additional discussion on potential human specific astrocyte subtype and gene expression differences would be helpful, particularly in light of several recent studies pointing to human specific enhanced complexity of astrocytes. 

Response. As suggested, we have added discussion to section 4.1, page 15-16.